# Salvaging Utopia: Lessons for (and from) the Left in Rivers Solomon's *An Unkindness of Ghosts* (2017), *The Deep* (2019), and *Sorrowland* (2021)

Megen de Bruin-Molé 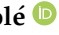

Winchester School of Art, University of Southampton, Park Avenue, Winchester SO23 8DL, UK; m.j.de-bruin-mole@soton.ac.uk

**Abstract:** In response to this special issue's question of whether mainstream science fiction has become stuck in presentism and apocalypticism, this article examines how utopia is expressed and salvaged in the work of Rivers Solomon. Using three of Solomon's novels and the theoretical lenses of black utopia studies and salvage-Marxism, I suggest that scholars and activists should approach this question from a different perspective. While Solomon's novels may seem dystopian from the perspective of liberalism or whiteness, they can also clearly be placed within the long, if marginalized, history of leftist and black utopian thought. Likewise, where the 'traditional' utopia (a concept I interrogate) is often imagined as grounded in hope and futurity, black utopia and salvage-Marxism reject these concepts as counterproductive to the actual work of social justice and utopia-building. Despite their presentism and apocalypticism, then, I argue Solomon's novels are very much utopian: they simply locate their utopian desire in radical kinship and salvage, rather than universalism or futurity.

**Keywords:** salvage; salvage-Marxism; salvagepunk; black utopia; utopia and dystopia; science fiction; desire; Afrofuturism; universalism; radical kinship





## 1. Introduction: Salvaging a Leftist Utopia

When asking ourselves whether we inhabit a world post-utopia, in which presentism and nostalgia reign, we must always add the caveat: utopia for whom? If globalisation and capitalism have indeed caused an "inability to imagine a future radically different from the present", and led science fiction to "abandon utopias in favour of post-apocalyptic narratives that presented the disaster as durational and never-ending rather than as a step toward a better (or at least different) future" as this special issue suggests (Gomel 2021), *whose* imaginings are implicated in this move? I am not disputing the turn in both the left and in mainstream science fiction towards the post-apocalyptic and post-utopian. Instead, I want to consider what we mean when we envision a 'better' or 'different' future in this context, and what implications this might have for the way we are approaching the problem of its presentism. This argument of this article is simple: utopian fictions are all around us, but to see the better worlds they present, we may need to adopt a different viewpoint. As Alex Zamalin writes, "utopian thought is endless, knows no ideology, and is, in the final analysis, a matter of perspective", including civil rights and social justice movements but also "contemporary right-wing faith in free markets and government deregulation" (Zamalin 2019, p. 5). While the right feeds and grows on its 'Retrotopias', the left seems paralysed to counter the right's nostalgia with an equally successful vision of the future. In a 2015 essay publicising her co-edited collection *Octavia's Brood*, a tribute to science fiction writer Octavia E. Butler, Walidah Imarisha suggests that for "all of our ability to analyze and critique, the left has become rooted in what *is*. We often forget to envision what *could be*" (Imarisha 2015). For Imarisha, salvage is a crucial part of rediscovering this vision; too often we "forget to mine the past for solutions that show us how we can exist in other

forms in the future" (Imarisha 2015). We need to salvage the past to save utopia, using what we can to move towards this end and discarding the rest.

To explore this process of mining or salvaging the past for leftist ends, I turn to scholars writing on black utopia and Afrofuturism. This is not the only lens we might use to salvage utopia, but it is an important one. Part of the reason, I would argue, for the seeming renaissance in utopias and so-called 'optimistic' science fiction is precisely because mainstream SF/F is seeing a rise in voices that do not represent the particular cis, white, male perspective that dominated the genre's vision (exceptions like Octavia E. Butler, Samuel R. Delany, and Ursula K. Le Guin notwithstanding) in the 1970s or 1980s. Not everyone has been stuck in presentism and nostalgia—only the loudest voices. Imarisha's proposed solution to the left's presentism, likewise, is a politicised and explicitly activist science fiction, paired with "the belief that all organizing is science fiction. When we talk about a world without prisons; a world without police violence; a world where everyone has food, clothing, shelter, quality education; a world free of white supremacy, patriarchy, capitalism, heterosexism; we are talking about a world that doesn't currently exist. But collectively dreaming up one that does means we can begin building it into existence" (Imarisha 2015). Crucially, however, this process also requires "that we see those who have been marginalized not as victims but as leaders and recognize that their ability to live outside acceptable systems is essential to creating new, just worlds" (Imarisha 2015). Black utopian stories and theories explore the ways capitalism, colonialism, and globalization ended worlds long ago but did not manage to extinguish a utopian vision—only to alter it. This writing prefigures more recent work in salvage utopia in important ways, and I echo Zamalin's hope "that combining black utopia's unseen transformative possibilities with an awareness of its limitations can invigorate contemporary political thinking" (Zamalin 2019, p. 2).

Informed by and in response to these kinds of conversations, many contemporary science fictions re-examine utopia's futurity in terms of salvage and leftist activism. Of these examples, Rivers Solomon's work is of particular interest because of how it has excited academic (and therefore largely white, middle class) but also leftist science fiction and activist communities over the last few years. Solomon's work is also highly productive for the way it explores the same utopian themes across different formats and genres, and from a variety of different perspectives. In this article, I approach Solomon's work as a set of heuristic examples, rather than offering extended close readings and background for each text. This is both for reasons of space, and to avoid further complicating the nuanced theoretical frameworks I aim to link to each other. Each of Solomon's novels is certainly worth its own, more detailed analysis in terms of its engagements with other utopian frameworks and themes of activism, race, gender, kinship, and labour—and indeed many such analyses are recently published or forthcoming. Here, however, I am most invested in thinking through the more general implications of using such texts as exemplars for leftist thinking and activism.

Solomon is a (queer, black, Jewish) American writer and activist currently living in Cambridge, UK. Like many SF/F writers working today, Solomon's stories often defy genre. *An Unkindness of Ghosts* (Solomon 2017), Solomon's debut novel, is the only straight-forwardly science fictional piece in faer[1] longform oeuvre, in that it is set in space. It details the journey of a plantation ship called the *Matilda*, which departed Earth long ago and still maintains the segregated, plantation-based structures of the world it left behind under the pretence of scarcity and survival. Solomon has also written a number of short stories, many for SF/F magazines like *Tor.com* but also for more mainstream US outlets like *The New York Times* and *Guernica*. *Sorrowland* (Solomon 2021), faer most recent novel, has been termed "Gothic" by publishers but also contains science fictional and posthumanist elements (Tor.com 2019). Solomon's 2019 novella *The Deep*, while set in a far future, explores the deep sea rather than outer space. This is a nod to both the Hugo-nominated 2017 single of the same name by hip-hop band clipping, with whom Solomon authored the novella, and

to Drexciya's electronic Afrofuturist albums of the 1990s and 2000s, from which both the single and the novella take their impetus.

Some of Solomon's writing could be situated within Afrofuturism, both because Solomon's protagonists are often black people (American or otherwise) striving to imagine a better world for themselves and their families, and because Solomon explicitly uses these fictions to address some of the struggles of the present. I would argue that faer writing fits more comfortably within the fictional tradition of black American SF/F writers like Samuel R. Delany or Octavia E. Butler, and indeed contemporary scholars have compared Solomon and Butler (see Ray 2019). Like Delany or Butler, Solomon's work is sometimes Afrofuturist, but it is often fantastical in a broader sense than this label implies. Solomon is also clearly writing in response to the late-20th-century shift towards 'social' science fiction (a tradition in which Ursula K. Le Guin's work also looms large). As Zamalin points out, however, "black utopia is irreducible to Afrofuturism, which has long been associated with science fiction and technology in the future, replete with robots and supercomputers. Indeed [ . . . ] if black utopia has been an expression of what the political scientist Richard Iton has called the 'black fantastic,' it has, as much as anything else, been a fantastical meditation on untapped possibilities already embedded within society—unconditional freedom, equality, interracial intimacy, solidarity, and social democracy" (Zamalin 2019, p. 10). It is productive to consider this irreducibility of black utopia when reading Solomon's work as well, not least because of how it moves across different mainstream genres that are less explicitly labelable as 'Afrofuturistic'.

At first glance we might also hesitate to label Solomon's work as utopian. At the time of writing, only *The Deep* had caused critics to use the term 'utopia', and then only to point out the ways the novella actually causes us to question the nature of the form (Wolfe and Mond 2020). Solomon faerself is ambiguous about the term, suggesting of *The Deep* that:

> It's utopian in the sense that the *wajinru* [the central society in the novella] occupy a realm that doesn't have to contend with artificial scarcity and hunger, social inequality, poverty, colonialism—capitalism. Under the sea, there are no landlords hoarding property so that the poor have nowhere to shelter. Food is not produced and commodified by landowners, unavailable to those who cannot afford it.

> But I don't think any community can ever be utopian, really, because part of the human project is contending with the fact that we are both individuals and social animals, and there will always be some dissonance between those two bodies. (Coleman 2019)

Teasing out the embodied dissonance Solomon describes, this article examines faer work through the lens of salvage-Marxism—a leftist perspective that aims to productively revisit and recuperate utopia, while also insisting that 'all is waste', that we are living in the aftermath of the apocalypse, and that there is no future (that it is easier to imagine the end of the world than the end of capitalism). It does so by attempting "a superposition and simultaneity of two traditionally counterposed strands within salvage-Marxism: a long-term 'institutional' counter-hegemonic strategy drawing on left-Eurocommunism [with its investment in civil rights and welfare], and a relentless antinomic left-Bolshevik insistence on rupture" and anarchy (Salvage Editorial Collective 2015a). These and related values of salvage-Marxism, as well as key themes from black futurists and utopian thinkers, can be illustrated through Solomon's writings.

## 2. Life after the End: Salvaging Utopia from Dystopia

Recent studies of black utopia explore how the genre has inevitably been built on a foundation of struggle and suffering. As Zamalin suggests, tracing US utopian vision from black nationalist Martin R. Delany to Afrofuturist Sun Ra and beyond: "Perhaps the reason scholars have inadequately explored the concept of utopia in the black American tradition is because much of black American life has been nothing short of dystopian" (Zamalin 2019, p. 6). Jayna Brown, likewise, suggests that "Dystopia, the horrific terms of being black in our earthly condition, is a starting point for critique; dystopia forms the terrain of

our existence" (Brown 2021, p. 1). It is from this space that black utopia is born, and where "forms of black life and liveliness are claimed and created in the terror". In black utopia, life does "not end with death, social or otherwise", for otherwise it could not exist at all, let alone flourish. Instead it exists "on the other side of death, where we reside" (Brown 2021, p. 1).

Solomon's stories also tend to begin as dystopias, imagining worlds in which the characters are cut off from the future by a grim and often fatal present. Speaking to Aster Grey, the protagonist of *An Unkindness of Ghosts*, a fellow passenger explains the perils of hopes and expectations on the *Matilda*:

> We got a saying here in Tide Wing: *Should* is for weaklings. Why would we care about such a thing when already nothing is how it should be on this cursed ship? *Should* won't make it so you don't got to cut off my foot, will it? It sure won't turn the heat back on, or kill the man who thought to turn it off in the first place. *Should* disappeared three hundred years ago when our old home went gone. There's no such thing as *supposed to* in space. (Solomon 2017, Chapter One, Loc 764)

The *Matilda*'s enslaved population has literally nowhere else to go but the cold recesses of space, and a revolution against their oppressors would spell their own demise. They are trapped in a cycle of crisis and present need.

In *The Deep*, there are multiple beginnings, from the memories of different historians of the *wajinru* people, and each is similarly grim. The *wajinru* are a civilization of water-breathing humanoids descended from enslaved black women who were thrown overboard during the Middle Passage. "What does it mean to be born of the dead?" asks the wajinru's very first historian, reflecting on their civilization's horrific origin. "What does it mean to begin? First, gray, murky darkness" (Solomon et al. 2019, p. 42). The *wajinru*'s most recent historian, Yetu, is the keeper of the people's ancestral memory, which is deemed too painful for individual *wajinru* to carry all the time, and is instead shared with the community through a regular Remembrance ceremony. The pain also proves too much for Yetu to bear, however, and she abandons her people in the middle of a Remembrance, leaving them overwhelmed and trapped in the past.

*Sorrowland* likewise begins after the end—personal rather than global—though here the apocalypse is a freeing force as well as a frightening one. Vern, teenaged, pregnant, and infected with a mysterious substance, has escaped from an unwanted marriage to the leader of the 'Cainite' black nationalist cult compound and must survive on her own in the woods. She is not alone; a malicious stranger haunts the woods with her. Vern is desperate, despondent, and "Ravenous. For what? For goddamn what? There was nothing in these woods but darkness and a fiend who killed not for food or hide but for the pleasure it arose in him to end the life of something small. She'd fled the compound in want of something, and though she'd been gone for only a short while, she already knew she'd never find it" (Solomon 2021, pp. 9–10).

How does one get from dystopia to utopia? Writing about salvagepunk (an aesthetic that precedes and informs salvage-Marxism), Evan Calder Williams discusses how the end of history that postmodernism signalled was, like all apocalypses, never really the end. Rather than simple rebellion (a rebellion he calls a kind of "apologist participation" on p. 30) against the historical metanarratives that have led us to this present reality, he suggests we must instead look for the alternate stories within those metanarratives. Most importantly, we must struggle against "current trendlines of nostalgia, the melancholia of buried history, and static mourning for radical antagonistic pasts seemingly absent from contemporary resistance to capitalism" (Calder Williams 2010, p. 20). Calder Williams does not condemn all modes of engaging with history, however, only advocating for a more antagonistic or uncomfortable stance towards it. Solomon's writing is likewise very interested in the possibilities opened up by the salvage of the past, not for the purpose of mourning or fetishizing (though space is given for these modes of engagement with the past as well), but for working with and through cultural memory. In the rest of this

article, I want to examine two seemingly contradictory themes in Solomon's writing that work to salvage utopia from dystopia, in which salvage-Marxism is also invested: utopian pessimism and radical kinship (a term I take from Miriyam Aouragh). Together, as in the arcs of Solomon's stories, these forces do not restore a universal utopian vision of the future—for arguably such a thing has never existed—but they open up a space in which the marginalized and defeated (the post-apocalyptic) can imagine the possibility of utopia once more.

### 3. Radical Kinship: Salvage-Marxism, Utopian Pessimism, and Strategies for Despite

In the inaugural issue of *Salvage Magazine*, published the same year as Imarisha's essay about *Octavia's Brood*, author and activist China Miéville writes that although utopias are "necessary", they are "insufficient: they can, in some iterations, be part of the ideology of the system, the bad totality that organises us, warms the skies, and condemns millions to peonage on garbage scree" (Miéville 2015, p. 181). *Salvage* is a collective of (mostly white) activists, authors, and artists in the UK, which aims to theorise a 'salvage-Marxism'. Though "made up of activists", much like Imarisha, *Salvage* sees its work as "not—yet, perhaps—an activist project, but an investigative one" (Salvage Editorial Collective 2015a). According to the journal's opening manifesto, "salvage-Marxism proceeds from the certainty that there is no certainty of capitalist collapse, and/but/because we live, already, in the landscape of such a collapse; of profits and superprofits sustained in a rubble of declining profit rate by undefeated and increasing exploitation, relative and absolute". In response, they propose a salvaging and repurposing of Marxist thought and theory: "From what we have inherited we keep what we can and reject what we must. We search history's dump to reclaim the best of what the Left has discarded" (Salvage Editorial Collective 2015a). In the manifesto, that which must be discarded includes hope (though crucially *not* joy), universalism, and futurity—things which cannot be reclaimed directly but might, through the rupturing and co-opting of their opposites, be salvaged.

For salvage-Marxism, the first and most important step towards salvaging utopia is to discard hope. This idea needs qualification, since salvage-Marxists are not suggesting we abandon all positive, progressive thought and feeling. Rather, salvage-Marxists "share uncertainties, questions, and a reading of many of the Left's traditions, including its 'optimism', as barriers to meaningful political action" (Salvage Editorial Collective 2015a). Hope has been co-opted as a means of discouraging action in the face of oppression. Miéville advocates "an end to one-nation apocalypse" and the idea that 'we are all in this together', instead offering a "hope that abjures the hope of those in power" (Miéville 2015, p. 189). Salvage-Marxism asks us to reframe and re-evaluate the utopian impulse through hope's opposites: pessimism, rage, and spite. In Miéville's words:

> There is hope. But for it to be real, and barbed, and tempered into a weapon, we cannot just default to it. We have to test it, subject it to the strain of appropriate near-despair. [ . . . ] We need utopia, but to try to think utopia, in this world, without rage, without fury, is an indulgence we can't afford. In the face of what is done, we cannot think utopia without hate". (Miéville 2015, p. 189)

In their second edition of the magazine, the *Salvage* editorial collective frames the situation as a post-apocalyptic one, in which sometimes "it seems hopeless, and yet, and yet. We go on, despite that" (Salvage Editorial Collective 2015b). This strategy is grounded in a utopian pessimism, which offers, in place of hope, "a strategy for despite" (Salvage Editorial Collective 2015b), "a *pre-emptive potential aftermath*. We have come to it after many failures, and now in guarded advance—but it is conditioned by the fact that we are always truly open to its other. For which 'hope' is too weak a formulation" (Salvage Editorial Collective 2015a).

Similarly, in many expressions and studies of black utopia, hope is dismissed as unproductive. As Brown writes:

> I don't think utopia needs hope at all. Hope yearns for a future. Instead, we
> dream in place, in situ, in medias res, in layers, in dimensional frequencies. The
> quality of being I find in the speculations considered here is about existence
> beyond life or death, about the ways we reach into the unknowable, outside
> the bounds of past, present, and future, of selfhood and other. This is what I
> call utopia: the moments when those of us untethered from the hope of rights,
> recognition, or redress here on earth celebrate ourselves as elements in a cosmic
> effluvium. (Brown 2021, p. 1)

Solomon's work thinks through utopia without hope on a number of levels. Faer
protagonists are marginalised, are often powerless, and are put through a great deal of pain.
Yetu and Vern spend most of *The Deep* and *Sorrowland* unable to change their situation,
resenting their communities, uncomfortable in their bodies, and suffering miscommunica-
tion and loss in their personal relationships. There is no one quite like them in the world,
and they experience isolation and self-loathing because of it. Despite many moments of
joy and humour in *An Unkindness of Ghosts*, Aster's fate is ultimately complex, conflicted,
and bittersweet, with the discovery of a whole new world but the loss of her family. Her
situation, it seems, is hopeless:

> Aster lay down in the black dirt, the granules cooler than the coolest sheets on
> Matilda. Sadness twisted up inside her, like a rope or maybe like a snake or
> maybe like a rosary. Whatever it was, this gangly sorrow, it had tied itself around
> Aster's vertebrae and would remain quite a long while. She felt sentimental. She
> felt superstitious. She felt like she could cry and catch her tears in a magic vial,
> pour the tears over Giselle's face, and resurrect her. But Aster was too dehydrated
> to weep, and even if she weren't, the water would do nothing but wet Giselle's
> dead, indifferent face, then evaporate. Repositioning Giselle's fingers so they
> were interlaced with her own, Aster rested beside her. Water was not good for
> such times as this, insubstantial as it was. But dirt, dirt would do. They were
> sheathed in it. (Solomon 2017, Kindle Locations 4808-4814)

Without the lenses of black utopia and salvage-Marxism it can be difficult to locate
utopia here. On a metatextual level, it can also be very difficult to imagine utopia without
hope's futurity. Without hope, what is left? At the same time, however, this kind of 'what
if' problem is what science fiction excels at. Talking about *The Left Hand of Darkness* (1969),
Ursula K. Le Guin has famously remarked that in the novel she "eliminated gender, to find
out what was left" (Cummins 1990, p. 84). After the loss of hope and universalism and
futurity, what is left in Solomon's utopias is (perhaps surprisingly) joy. From a salvage-
Marxist perspective, this is only logical: "To earn its—real—pessimism, salvage-Marxism
is always-already surprised by joy"(Salvage Editorial Collective 2015a). In its 'utopian
pessimism', black utopia can be ironically antithetical to Afropessimism, which expresses
the impossibility of overcoming anti-blackness, foreclosing the possibility of meaningful
cross-racial solidarity (see Olaloku-Teriba 2018; Okoth 2020; Wekker 2021). Miriyam
Aouragh suggests that "what we want is a radical alternative based on kinship, a kinship
of equality; a universalism grounded in resistance" (Aouragh 2019, p. 21). For this reason,
it is "key to repeat that there is not one black or one white epistemology; our political
differences are in essence ideological and not biological" (Aouragh 2019, p. 16).

From the perspective of radical cross-racial solidarity, if we are to draw a comparison
once again between Solomon and Butler, there is a clear effort in both authors' writings to
soften their utopian pessimism to a degree where it is relatable to a mainstream audience
(though in the case of Butler's historically-minded fantastical novel *Kindred*, it is interesting
to note that "she repeatedly referred to it instead as a 'grim fantasy' [rather than science
fiction] and asked her publisher to do the same") (Canavan 2016, pp. 19, 56). This does
not mean these texts are never blunt or harsh. Solomon's work, including *An Unkindness
of Ghosts*, has been described as violent, grim, and unpleasant in reviews, and it can be
these things at times. I would suggest, however, that despite these moments (or, from a
salvage-Marxist perspective, because of these moments), *An Unkindness of Ghosts* is wildly

optimistic. While it does not deign to soften its narrator's experiences of violence and marginalisation for white readers, it still imagines the utopian possibility of allyship and friendship that cuts across gender, racial, and other boundaries. More so, it finds joy in these relationships, distinct from but alongside the moments of black joy throughout the novel. This is not without qualifiers and reservation, but it is also far from a given in our real world. As Zamalin notes, "reflections upon black utopia were defined by a unique take on perennial questions of political theory [ . . . and] combined this understanding with what realists rejected—belief in the collective popular will, a defense of equality, and a defense of localized knowledge" (Zamalin 2019, p. 13). This may seem like a simple move, but I would suggest that here Solomon's work again parallels Butler's, in that it is deliberately less pessimistic about the situation that it could be. As Gerry Canavan writes of Butler's fiction:

> In the archives—in the scenes Butler cuts from the novels, in the work she abandoned, in the stories she outlined but never began—we can see distinctly the lifelong tension between Butler's desire to write what she always called a YES-BOOK (a universally loved bestseller) versus the NO stories she felt driven to tell despite her ambition for sales and for fame. The Huntington archives make clear that Butler typically edited her writing in order to make it more optimistic. The drafts, consistently, are much more brutal and unforgiving than the published works, often filled with much more extreme violence and resulting in unhappier outcomes (a surprising fact, given how gruesome and disturbing the published books can be). (Canavan 2016, p. 19)

Though surely indebted to Butler's work—as are huge swathes of contemporary SF/F and utopian writing—Solomon is part of a very different generation of writers, working at a time when black SF/F is arguably enjoying greater public attention (and sales) than ever before, and where 'gruesome and disturbing' experiences of marginalised peoples are beginning to become more speakable in a mainstream context. Faer descriptions of black pain (and black joy) should be read within this context. However, for the marginalised, allyship is still by no means a given. As one reviewer notes of *An Unkindness of Ghosts*, the "cross-class, cross-racial, wonderfully queer relationship" between Aster and Theo makes cooperation seem, if not easy, then surprisingly achievable: "where another author might exploit these communication glitches to drive a dramatic wedge between them, Solomon demonstrates the ease of repair" (Milks 2018). Theirs is not a rational bond—it is a radical one. It requires work and a shared vision. Aster and Theo can maintain their relationship through every hardship because they share a common goal and a common enemy: in Aouragh's words, speaking of black-Palestinian activism, "solidarity has grown because both understand the relationship between state violence, the prison system, and militarised surveillance" (Aouragh 2019, p. 21). This solidarity in turn opens a space where vulnerability, growth, and action are possible. Likewise, salvage-Marxism's second, related strategy for rediscovering utopia is radical kinship, despite (and because of) agendas that are placed into competition with each other by capitalism and austerity: "we hold that there are specificities to contemporary capitalism that have a savage effect on the grounds for mass, rebel, class consciousness, simultaneously as they make their own destruction ever more necessary" (Salvage Editorial Collective 2015a). In *An Unkindness of Ghosts*, as in salvage-Marxism, recognition of difference is not enough: "Worse, it is a poor substitute for an admittedly more difficult task—activism invested in taking on structural inequality through the tough work of coalition building" (Aouragh 2019, p. 17). It is the desire for and joy in that salvage and coalition-building work, not the hope of its future acknowledgement, that needs to be repeatedly and persistently imagined. This is the task Solomon's work also attempts: to imagine and dramatize the work of solidarity, in all its specificity and nuance, but also in its joy.

In place of hope for the future, as Jayna Brown theorises utopia she introduces desire for other temporalities and experiences of the present, arguing that we "can think of desire, and especially its fulfillment, as deeply political in the context of black life" (Brown 2021, p. 14). In the case of black utopia, and as embodied in expressions of black joy: "Utopia is a state of being and doing" (Brown 2021, p. 1). This is not the same as hope, or as a universalising optimism. It is quite the opposite: recognising joy and desire in one's body despite the dystopian forces that work to eradicate it, and desiring love, solidarity, and kinship with others *in spite of* the obstacles of our reality (in both senses of 'in spite'). This is a subtle but important difference with how joy is often conceptualised in other frameworks. It is a politics that refuses universalism and the flattening of black identity, acknowledging that the work of anti-racism can at times "be overwhelming", but also imagining and building a world where leftist "movements, workplace disputes and strikes, campus sit-ins and demonstrations are sites of resistance where we breathe, where hope overtakes fatalism. Where our scepticism is not frivolous, but sober. And where we learn and unlearn" (Aouragh 2019, p. 22). In terms of learning and unlearning utopia, Brown suggests that we might "think of desire differently: not as consumption but as relational and charged with the potential to explode all attempts to order and contain it" (Brown 2021, p. 14). She cites Herbert Marcuse's figuration of utopia as a "qualitative change in the character of wants and needs" (Levitas 1990, p. 160; in Brown 2021, p. 14), suggesting that our task is to "teach desire to desire, to desire better, to desire more, and above all to desire in a different way" (Thompson 1977, pp. 790–91; in Brown 2021, p. 14).

Solomon's novels represent a similar logic. *The Deep* offers what is perhaps the most joyful, conventionally utopian ending. Though Yetu leaves the community of *wajinru*, she finds a human one that helps her to see value in shared history, however painful. In the end, "instead of taking the History from them, she could join them as they experienced it [ . . . ] they would bear it all together" (Solomon et al. 2019, p. 148). Crucially, this is not a universalising oneness—each *wajinru* takes on different parts of the History in unique ways, and the accompanying mess and struggle is acknowledged and important. Yetu acknowledges her people as individuals and collective but also commits fully to the work of building connections to and among them, giving "her whole being to the ocean the way the ocean had given all of itself to her, giving the *wajinru* the spark of life, showing them that if only they knew how, water could be breathed" (Solomon et al. 2019, pp. 143–44). This gift does not only extend to Yetu's people, but to everyone who takes part in radical kinship. Just as the first *wajinru* were able to survive the brutalities of the Middle Passage by learning to breathe in the sea (as they did in the womb), so Oori, the human Yetu grows to love, is transformed into "a completely new thing", neither human or *wajinru*, and joins Yetu in the deep (Solomon et al. 2019, p. 155).

In *An Unkindness of Ghosts*, kinship represents a grimmer, if no less utopian, struggle. In a violently (self-)harmful act, Giselle burns Aster's laboratory, destroying medicines that benefit the enslaved passengers of the *Matilda*, but also the maps and notes that might have offered them a future among the stars. Despite this, Aster commits to Giselle, and to the larger goal of escaping the *Matilda* that she and Giselle have fought for: "As much as it frustrated her, she understood the logic of Giselle's psychosis. Everything dies, so exert control by burning it away yourself. Everything will be born again anyway. There's no such thing as creation, merely a shuffling of parts. All birth is rebirth in disguise" (Solomon 2017, Kindle loc 809). Aster's commitment to this goal and this relationship, in spite of Giselle's violence and restlessness, ultimately lead Aster to the paradise of the new Earth, where they arrive (too late) together.

At the end of *Sorrowland*, Vern has reimagined utopia entirely. After repeated loss, pain, and monstrous transformation, she at last finds joy and kinship: with her children, with Gogo—the Lakota woman who fights with and for her in spite of hardship and misunderstanding—and in the darkness, desperation, and liveliness of the once-dystopian wood:

Vern smiled at all the loves of her life. There was Howling. There was Feral. Now there was Gogo, too. She almost cried, so grateful she was. "You okay?" asked Gogo. Vern nodded and wiped away the single tear threatening to fall. "I like the woods", she said. "In them, the possibilities seem endless. They are where wild things are, and I like to think the wild always wins. In the woods, it doesn't matter that there is no patch of earth that has not known bone, known blood, known rot. It feeds from that. It grows the trees. The mushrooms. It turns sorrows into flowers".

They both sat down, sweaty arm to sweaty arm. They remained until the woods were black but for patches of moonlight. They remained until they could hear the night calls of one thousand living things, screaming their existence, assuring the world of their survival. Vern screamed back. (Solomon 2021, pp. 354–55)

Ultimately, salvage-Marxism and black utopia are also about salvaging desire *in spite*: for ourselves, and for others, in defence of radical kinship under capitalism.

## 4. Conclusions: Finding Out What Is Left

In *Sorrowland*, Solomon quotes Ursula K. Le Guin's 1983 "Left-Handed" commencement address (Le Guin 1989) in a passage that is worth reproducing here in its entirety:

*"I hope you live without the need to dominate, and without the need to be dominated", Gogo read, pausing to ensure she had Vern's full attention. "I hope you are never victims, but I hope you have no power over other people". Gogo's voice, crisp, dulcet, and deep, seemed made for oration. Vern always sprang to alert at the sound of it. "And when you fail, and are defeated, and in pain, and in the dark", Gogo read on, "then I hope you will remember that darkness is your country, where you live, where no wars are fought and no wars are won, but where the future is".*

It was novel to feel moved by a stranger's words, and Vern regarded this rousing feeling with suspicion. Was this what it was like to be a Cainite? To hear the words of a sage and actually believe them? To find in their message a small truth?

Vern had always loved stories, but years of listening to Sherman disguise lies with rhetoric on the pulpit had made her less generous in her attitude about writing.

"Go on", said Vern, cautiously curious.

Gogo turned the page. *"Our roots are in the dark; the earth is our country. Why did we look up for blessing—instead of around, and down? What hope we have lies there. Not in the sky full of orbiting spy-eyes and weaponry, but in the earth we have looked down upon. Not from above, but from below. Not in the light that blinds, but in the dark that nourishes, where human beings grow human souls"*, said Gogo, finishing. Darkness was Vern's country. It was all she had by way of a homeland. It pleased her to think it could be a place that nourished. Everyone was always going on about light this and light that, but what of dark? (Solomon 2021, pp. 212–13)

In this re-reading of Le Guin, just as in Imarisha's essay, the future of the left comes "from below", from the resilience and resourcefulness of the marginalised, and from the nourishment to be found *in spite* of darkness—in celebration of darkness. From this perspective of hope in the darkness, even the ending of *An Unkindness of Ghosts*, with Aster and Giselle "sheathed" in dirt can be read as utopian. The reader does not know *how* Aster will go on, but at the same time her arc in the novel has taught us that she will do just that: that darkness is her country, where she lives and finds nourishment *in spite of* her oppressors, and where the future of humankind lies as a result.

Black utopia and salvage-Marxism reject the false universalism of hope, and as a result they do not always offer a happy ending. All they can promise is an open horizon, and a "strategy for despite" (Salvage Editorial Collective 2015b). For Zamalin, "That black utopia was often left unelaborated was less a failure of imagination and more a defense of keeping alive a horizon, which would exist as unfulfilled possibility" (Zamalin 2019, p. 14). Salvage-Marxism, likewise, promises that it will continue to "seek to understand

and plan what we must do, what we already do: *Salvage*, like countless activists, like all the clear-eyed Left, goes on in the face of outrages, despite capitalism's despite [ . . . ] a socialism that can survive the contempt and spite of the rulers, and that weaponises its own back at them" (Salvage Editorial Collective 2015b). In *An Unkindness of Ghosts*, *The Deep*, and *Sorrowland*, this is presented as a radical leftist utopia worth salvaging.

**Funding:** This research received no external funding.

**Acknowledgments:** I am indebted to the editor of this special issue, Elana Gomel, for her invaluable feedback and support during the writing process, *in spite* of a global pandemic. I would also like to thank the three anonymous reviewers of this article for their careful reading and criticism, which allowed me to see the piece through new eyes and to clarify the argument in important ways.

**Conflicts of Interest:** The author declares no conflict of interest. The funders had no role in the design of the study; in the collection, analyses, or interpretation of data; in the writing of the manuscript, or in the decision to publish the results.

## Note

[1] Solomon is nonbinary and uses the pronouns fae/faer. Though outside the scope of this article, from a utopian and feminist science fiction perspective it is important to note that Solomon also plays regularly with gender in faer fictions. For instance, despite the familiar pronouns used for the protagonists in faer novels (always she/her), none of these characters embody traditional, binary gender identities. This article uses the same pronouns for these characters that are used in Solomon's texts.

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
