# Peer review of "Salvaging Utopia: Lessons for (and from) the Left in Rivers Solomon’s An Unkindness of Ghosts (2017), The Deep (2019), and Sorrowland (2021)"

_humanities, doi:10.3390/h10040109_

Round 1

Reviewer 1 Report

This is a fascinating, extremely clearly argued and convincing paper. It fully fits the remit of the special issue, and provides an important addition to thinking about Black Utopianism. The reading of River Solomon’s work within — and against — frames of salvage-Marxism and Afrocentrism is innovative, and makes a strong case for reimagining the futurism of utopian thought. The argument is polemical, in that it is challenging established conceptualisation of the presentism and dystopian rhetoric in Black arts; but this is not merely a rhetoric ploy. The case for thinking about grounded social change, a Utopianism in the face of the precarity and desperation of late Capitalism, is both important to contemporary thinking about racism and it is extremely well-made and supported by the example of Solomon’s work.

This is an excellent paper!

I noticed a few typos, for correction as follows:

p. 2, l. 64: change ‘black’ to ‘Black’

p. 3, l. 127: change ‘we living’ to ‘we are living’

p. 6, l. 260: change ‘protaginists’ to ‘protagonists’

p. 8, l. 370: change ‘In the end An Unkindness of Ghosts kinship represents a grimmer’ to ‘In the end, kinship in An Unkindness of Ghosts represents a grimmer,’

p. 8, l. 399: change ‘Ultimately, then salvage-Marxism’ to ‘Ultimately, then, salvage-Marxism’ 

Author Response

Thank you! Both for your generous review (which made my day) and for your careful reading and spotting of typos. I have corrected these in the final version.

Reviewer 2 Report

Some misprints need to be corrected: 

  • l. 64, black utopia: Black utopia
  • l. 127, that we living: that  we are living
  • l. 260, protaginists:  protagonists
  • l. 334, as Jayna Brown theorises utopia she introduces: as Jayna Brown theorises utopia, she introduces

Author Response

Thank you! I have corrected the typos you note in the final version.

Reviewer 3 Report

Comments on “Salvaging Utopia: Lessons for (and from) the Left in Rivers Solomon’s An Unkindness of Ghosts (2017), The Deep (2019), and Sorrowland (2021)”

In my opinion, the essay is simply too cursory (at points) and has not done enough to paint a descriptive and interpretive picture of Solomon’s work. Although this topic will surely be an important contribution to the special issue, in its current form, the author has only alluded to topics and ideas, but could take more time to develop them. The virtue of this piece is that it might point readers to relevant works of secondary literature, but it would have to develop its own interconnections, in my opinion, to provide readers with a convincing new window into Solomon’s important work and to be a foundational secondary text on Solomon.

The author provides a thesis statement about how Solomon’s work aims to productively revisit and recuperate utopia, while insisting that all is waste and that we are living in the aftermath of the apocalypse. Overall, what would be useful here is a more thoroughgoing exploration of some of the events in the novels in question, which are only loosely touched on in this relatively short essay. A reader doesn’t get a clear picture of how Solomon paints this world, and specifically (and more important) why it's all that different from other dystopias we have seen. They are different indeed, but not enough of that is coming across here.

Considering some of those other dystopias: I felt that the author might do well to differentiate Solomon’s utopias from the dystopias that are part of the science fiction canon, if for no other reason than they sound like they share much in common with that canon. To name a couple of examples that quickly come to mind: Godard's Alphaville is a film in which one can little imagine any future that doesn’t involve capitalism (or, at least, the status quo), and films like Mad Max or even Soylent Green are about literal salvage and waste. The recent Mad Max film (Fury Road) also posits the creation of a network of allies, etc. What is going on in Solomon's work that distinguishes it from other dystopias or utopian depictions? The author is arguably doing a disservice here to Solomon by not making clear what distinguishes Solomon's work from other dystopias.

Interestingly, the author of this essay develops this question of “salvage” -- specifically that what happens in the novels is not (necessarily) about salvaging of the junk and detritus of capitalism, but rather about salvaging the ideas of left, its plans and hopes. What remains for us is only a capitalism in which we can envision nothing but capitalism’s eventual continuation until we are saved from capitalism by the apocalypse. That’s all great material and was engaged with nearly every quotation attributed to the Salvage Editorial Collective.

On the other hand, I have to say that I was also somewhat unconvinced by the idea of joy as a distinguishing factor. Characters in many novels experience joy (even in dystopian novels) and I didn’t see how that is connected to the network of meanings being produced here (see esp. p. 6). Yet another avenue that it is opened but unpursued was Octavia Butler's language of “grim fantasy.” The author could, in my opinion, have done more to explain why that term doesn't do the work that he or she wants it to? I see that Solomon has been described as post-Butlerian – is this a good designation (along with posthuman, postmodern, and postapocalyptic)? Why or why not?

Moreover, as it is described, the contention that allyship and friendship in the novels cuts across gender, racial, and other boundaries makes this argument point seem smaller (more trivial) than it might be. Any utopian representation is going to entail possibilities of allyship and friendship. The specific nuances of these allyships and what differentiates them from other ones would be elicited through some of the missing close readings. The architecture of the close readings had been so little built that there's not much basis for one to make a judgment. Why don’t these key terms (joy and allyship in particular) simply add up to merely redemptive utopian novel?

Also, one question: the author mentions that Solomon is Jewish on page 2, but this never comes up again in the course of the essay. Was that observation going somewhere? Does Solomon ever speak about that in the many interviews?

Throughout this essay, the best passages are the ones that build on the ideas from the Salvage Editorial Collective, but I would like to have been better convinced that Solomon's novels correspond to the paradigm being presented to us, and that Solomon could be considered a representative novelist of salvage Marxism. There's a lot of room to do that here, if the author is moved to do so. Overall, this is a really interesting topic and – perhaps, apart from the undeveloped mentions of “joy” – an interesting constellation of ideas. I very much hope that the author develops it, and would, in my opinion, need to do so, in order to make this a contribution to research on Rivers Solomon.

Author Response

Thank you very much for this extended review. I deeply appreciated the time and care evident in your feedback, and though I was not able to incorporate all of your suggestions in the revised article, they have helped me to reframe the piece more effectively. 

While they are well worth exploring and analysing in more depth, to engage with all of the suggestions outlined in this feedback would require a much longer article (or indeed a book, which I someday hope to finish!). And as the editor of the special issue indicates in her notes, the article is less interested in proposing Solomon as "a representative novelist of salvage Marxism", and more in using Solomon's work as an exemplar to introduce the ideas of salvage-Marxism. Both to explain why these ideas (by themselves quite dense and complex) might help reframe the question raised by the special issue, and to link these ideas to ongoing work in black Utopia and Afrofuturism, where for instance the concepts of allyship and joy have a much different resonance than they often do in utopia/dystopia studies, or in white mainstream discourse. I have worked to signpost these points more clearly and consistently in the article, which I hope is now more successful in addressing some of the concerns raised, and in rationalising the absence of a more extensive close reading of each of Solomon’s novels.

The point that more could be said about Solomon's relation to Butler is well made, and I have added some commentary on this front in both the introduction and later sections that hopefully also helps to answer the question of why I don't consider Solomon's 'utopias' as part of the dystopian legacy. 

Likewise, I found the following question very helpful in thinking through my decision to describe Solomon as "a (black, Jewish) American writer": "the author mentions that Solomon is Jewish on page 2, but this never comes up again in the course of the essay. Was that observation going somewhere? Does Solomon ever speak about that in the many interviews?" Solomon speaks about faer experiences of marginalisation in different ways across different interviews. Because of the frameworks I am working with in this article I largely focus on blackness. Proceeding from the paragraph above this one, where Imarisha highlights how important it is “that we see those who have been marginalized not as victims but as leaders and recognize that their ability to live outside acceptable systems is essential to creating new, just worlds” (Imarisha 2015), for me the point of these brackets was to indicate identifiers that are potentially significant to my presentation of Solomon as a marginalised writer of activist SF/F. I also mention in a footnote that Solomon is nonbinary—though after considering this feedback I have added to the brackets: "(queer, black, Jewish)" to hopefully more clearly capture the marginalised identities Solomon most frequently identifies with.

Round 2

Reviewer 3 Report

The response to comments was very thoughtful. I'm glad to see that the author benefited from the editorial process, and I am confident that this will be an important new essay for readers interested in Rivers Solomon and for other readers of the special issue.